# Plasmonics Induced Multifunction Optical Device via Hoof-Shaped Subwavelength Structure

**Kui Wen, Zhaojian Zhang, Xinpeng Jiang, Jie He and Junbo Yang ***

Center of Material Science, National University of Defense Technology, Changsha 410073, China;
kuiwen93@hotmail.com (K.W.); 376824388@alumni.sjtu.edu.cn (Z.Z.); jackson97666@163.com (X.J.);
18795898068@163.com (J.H.)
* Correspondence: yangjunbo@nudt.edu.cn

**Abstract:** The electromagnetic spectrum includes the frequency range (spectrum) of electromagnetic radiation and its corresponding wavelength and energy. Due to the unique properties of different frequency ranges of the electromagnetic spectrum, a series of functional devices working in each frequency rang have been proposed. Here, we propose a periodic subwavelength hoof-shaped structure array, which contains a variety of geometric configurations, including U-shaped and rectangle structures. The results show that the enhanced optical transmission (EOT) effect of the surface plasmon excited by the hoof-shaped structure is highly sensitive to the polarization of the incident light, which leads to the peak's location shift and the amplitude intensity variety of transmission peaks of U-shaped structure in the case of coupling based on the surface plasmon of rectangle structure. In addition, take advantage of the EOT effect realized in the periodic hoof-shaped structure array, we propose a multifunctional plasmon optical device in the infrared range. By adjusting the polarization angle of the incident light, the functions of the optical splitter in the near-infrared range and the optical switch in the mid-infrared range are realized. Moreover, with the changes of the polarization angle, different proportions of optical intensities split are realized. The device has theoretically confirmed the feasibility of designing multifunctional integrated devices through a hoof-shaped-based metamaterial nanostructure, which provides a broad prospect for the extensive use of multiple physical mechanisms in the future to achieve numerous functions in simple nanostructures.

**Keywords:** plasmonic optical device; enhanced optical transmission; interaction between light and metallic film; coupled mode theory

## 1. Introduction

Surface plasmons (SP) include surface plasmon polaritons (SPPs) and local surface plasmons (LSP), which are surface electromagnetic waves formed by the collective oscillation of free electrons in a metal and the incident light field [1–3]. In recent years, the enhanced optical transmission (EOT) effect brought by metal-based surface plasmons has been applied in different wavelengths, including visible light, near-infrared and infrared ranges. Correspondingly, a large number of micro/nano structures have been proposed, such as: circle holes [4], stripe grating hole [5], single/double aperture hole [6], composite structure [7], cross-shaped hole [8], triangular hole [9]; Also, a large number of devices and applications have been constructed, for example: A absorber device that realizes broadband multifunctional properties by introducing vanadium dioxide into a metamaterial [10]; A filter device that realizes multiple channels working simultaneously by changing the number of concentric apertures [11]; A color sensor was realized using interference effects in the metal-insulator-metal Fabry-Perot (FP) cavity of polydimethylsiloxane (PDMS) as the dielectric layer [12]; The function of displaying different colors in the visible light range is realized by changing the material characteristics and the geometric parameters

of the structure, in order to achieve the application of the structure color [13–19]. However, the devices proposed by researchers have the defects of single function, complex structure and expensive materials; they cannot achieve better performance when the devices are integrated.

In this paper, the photoelectric effect of the polarization direction of the incident light on the periodic hoof-shaped subwavelength structure array is studied by numerical simulation. The results show that as the polarization angle of incident light changes, the transmission peak splits in the near-infrared range, and a new transmission peak appears in the mid-infrared range. By comparing the EOT phenomenon of three structures, it is found that in the hoof-shaped structure, the LSP resonance modes excited by the U-shaped and rectangle portions, respectively, are different from those when they exist alone.

Finally, we propose a multifunctional device work on different wave ranges with adjustable performance. By changing the polarization angle of the incident light, a mid-infrared range plasmonic optical switch with a contrast ratio exceeding 13.2 dB can be realized. Not only that, the device implements a plasmonic optical splitter in the near-infrared range, which can achieve optical splitting functions with different intensities.

## 2. Design and Modeling

Figure 1 is a cross-sectional diagram of periodic hoof-shaped structure arrays and a unit cross-sectional diagram, where the gray part is quartz substrate and the blue part is silver (Ag). $P_x$ and $P_y$ present the arrays periods along $X$ and $Y$ axes, respectively. The silver film thickness is 50 nm, which is placed on a quartz substrate with a thickness of 225 nm. The incident light (Gauss light source), the center wavelength is 1550 nm and the pulse width is 5 fs, is incident perpendicularly from one side of the substrate. Based on the 3D-FDTD method, we use MATLAB software to simulate the propagation of light in periodic subwavelength metal holes [20,21]. In order to ensure the accuracy of simulation, we set the step size in $X$, $Y$, $Z$ direction to 5 nm. The EOT of subwavelength hole arrays in metal film can be characterized by the transmittance $T$ as following [22]: $T = P_{out}(\lambda)/P_{in}(\lambda)$, where $P_{in(out)}$ is power flux through the metal film.

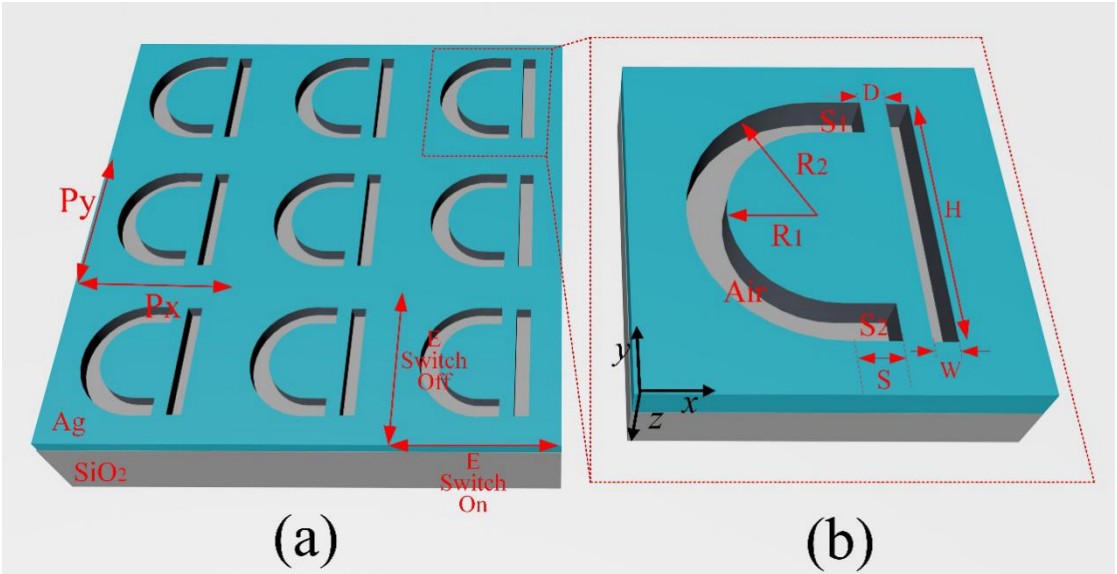

**Figure 1.** (**a**) Cross-sectional diagrams of the periodic hoof-shaped structure arrays. (**b**) Structural cross-sectional diagram of the unit structure in the *X-Y* plane. $R_1$ represents the U-shaped inner radius, $R_2$ represents the U-shaped outer radius; $W$ and $H$ are the width and length of rectangle; $D$ is the distance between two geometric shapes; $S$ is the length of the U-shaped tip. As follows: $R_1 = 100$ nm; $R_2 = 200$ nm; $W = 50$ nm; $H = 400$ nm; $S = 50$ nm; $D = 50$ nm; $P_x = P_y = 600$ nm; The incident light propagates in -$Z$ direction.

## 3. Results and Discussions

By considering the incident light polarization sensitive of structure [23,24], we altered the incident light polarization angles $\theta$, that is, rotated from parallel to Y axes to perpendicular to Y axes. Figure 2 shows the electric field distribution and transmittance when the $\theta$ changes. The surface plasmon resonance was excited in the U-shaped structure when $\theta$ is 0, which is distributed in two tips of the U-shaped structure in Figure 2a; With the increase of the polarization angle $\theta$, the charge in the tip gradually moved towards the middle. The charges converged completely in the middle of the U-shaped structure, when $\theta$ is 90. The difference was that, the localized surface plasmon resonance was not excited in the rectangle structure when the $\theta$ is 90, but with the increase of the angle, a large number of charges accumulate in the rectangle structure, resulting in the increase of the resonance intensity. To date, we find that with the increase of the angle $\theta$, the transmission peak in the near-infrared range splits. Moreover, a new transmission peak is generated in the mid-infrared range in Figure 2b.

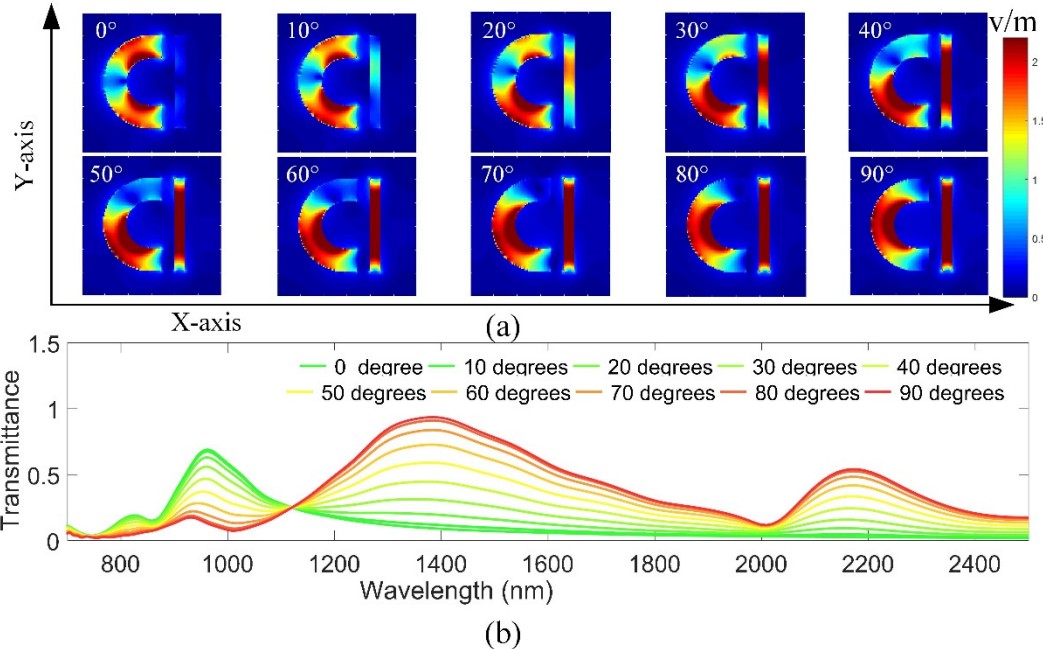

**Figure 2.** (**a**) Spatial distributions of the electric near field $E$ with the different incident polarization angles $\theta$ on the plane of $Z$ = 375 nm (detection plane), respectively. (**b**) Transmittance spectrum of hoof-shaped structure arrays with different $\theta$ with $P_x = P_y$ = 600 nm.

In order to demonstrate the mechanism of this phenomenon, we compared the EOT phenomena of three structures. As shown as in Figure 3a: In the U-shaped structure, with the increase of the angle $\theta$, peak I decreased and peak III increases in the near-infrared range; In the rectangle structure, peak II increases with the increase of the angle $\theta$; The important is that, all the transmission peaks have not appeared to shift. That is to say, in the separate U-shaped structure and rectangle structure, the change of angle does not cause the shift of transmission peak. Combined with the electric field distribution diagram in Figure 2a, we can see that: with the increase of $\theta$, the charge distributed in the tip of U-shaped structure gradually converges in the middle, causing the peak I decrease and the peak III increase. Therefore, the charge accumulated at the tip leads to the generation of transmission peak I, and the charge converges in the middle, which leads to the formation of peak III. Likewise, a large number of charges are excited in the rectangle, leading to the electric field increase, thus forming the peak II. But, in the hoof-shaped structure, as the electric field increases with the increase of electric charge in the rectangle, the U part is coupled with the resonance mode (LSP resonance) in the rectangle part, and then the Peak II and III shifts. However, it should be noted that the Peak I does not disappear completely when the angle is 90 degrees.

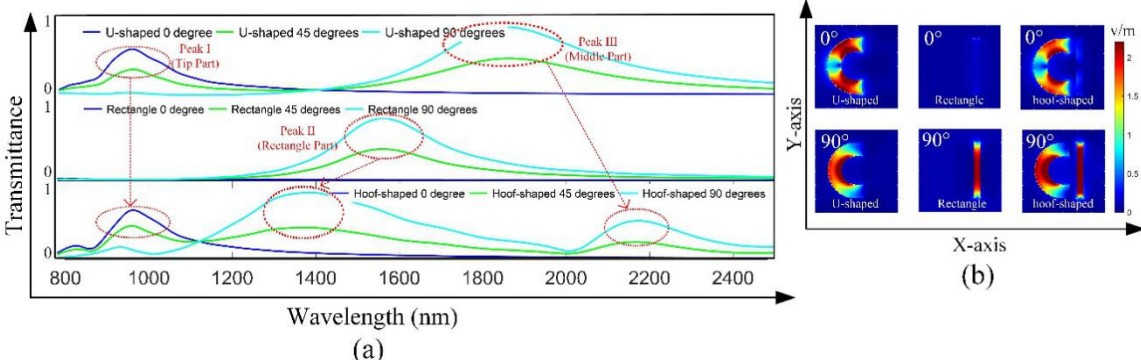

**Figure 3.** (**a**) Transmittance spectrums of the three structures (U-shaped, rectangle and hoof-shaped) under the different angles of the incident polarization, the structural parameters of the three structures remain unchanged. (**b**) Electric field distributions of U-shaped structure, rectangle structure and hoof-shaped structure at 0 and 90 degrees.

As shown in Figure 3b, in the case of 0 degree, since the rectangle structure does not excite the LSP resonance, the electric field distribution of U-shaped structure is almost the same as that of hoof-shaped structure. This shows that the transmittance of U-shaped structure and hoof-shaped structure are basically the same in different wavelengths at 0 degree. When the angle is 90 degrees, we can see that the relative electric field intensity in the U part of the hoof-shaped structure is greater than that in the U-shaped structure alone. In other words, due to the addition of the rectangle structure, the effective refractive index $n_{eff}$ in the U part increases [25,26], thus causing the redshift of the Peak III. It cannot be ignored that the excitation of the LSP resonance mode in the rectangle also makes the charge in the tip of the U part slightly increase, which is also the reason why the Peak I does not disappear completely. On the other hand, as mentioned in the literature [27,28], strong coupling will lead to the splitting, redshift and blueshift of the transmission peak. Therefore, with the increase of the angle, the charge in the U part gradually converges in the middle, and the intercoupling effect with the rectangle changes, which makes the Peak II blueshifts.

## 4. Plasmonic Optical Devices Based on Periodic Hoof-Shaped Arrays

### 4.1. Design of Plasmonic Optical Switch

Considering the induction of new transmission peaks in the mid-infrared range, we constructed a plasmonic optical switch (PO switch) by changing the polarization angle of the incident light. The parameters of the PO switch are shown in Figure 1b. In this paper, we consider that the center wavelength $\lambda_0$ of the transmission peak is the working wavelength of the PO switch and the performance index of the contrast ratio is used to quantitatively describe the control effect of the PO switch on the lamp [29]:

$$\eta = 10\log_{10}\left(\frac{T_{on}}{T_{off}}\right) \tag{1}$$

where $T_{on/off}$ is the transmission peak at the working wavelength $\lambda_0$ of the switch when the switch is on/off.

Furthermore, we change the length of $S_1$ in the hoof-shaped structure for break the structure's symmetry in $X$ direction. The parameters are as follows: $S_1$ is set to 0 nm, 10 nm, 20 nm, 30 nm, 40 nm, 50 nm and other parameters remain unchanged. As shown in Figure 4a, the redshift of transmission peak occurs with the increase of $S_1$, and the maximum redshift is 167 nm at $\theta$ is 90. It is apparent that, when $S_1$ is 50 nm, the maximum contrast ratio is 13.2 dB in Figure 4b. The results show that the symmetry of the structure is more complete with the increase of $S_1$. Generally speaking, when the two resonance modes are asymmetric in the Fano resonance (Resonance mode must be generated or the resonance intensity must reach a certain threshold), the transmission peak display splitting [30–32].

In other words, when the resonance modes of the two tips are asymmetric, the transmission peak will shift. At the same time, with the increase of $S_1$, the space in the whole hole increases, which leads to the increase of accumulated charge and the increase of transmittance.

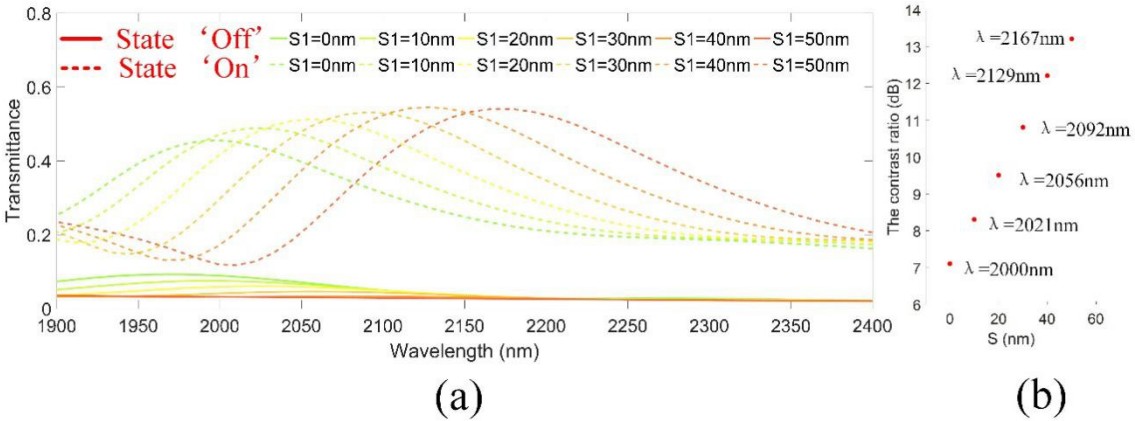

**Figure 4.** (**a**) Transmittance spectrum of changing the tip length $S_1$ in U-shaped structure; (**b**) The contrast ratio and working wavelength of plasmonic optical switch under different length $S_1$.

We change the length of distance $D$ in the hoof-shaped structure to study the influence of the EOT. The parameters are as follows: $D$ is set to 10 nm, 20 nm, 30 nm, 40 nm, 50 nm, 60 nm, 70 nm, 80 nm, 90 nm, and other parameters remain unchanged. As shown in Figure 5a, the transmission peak occurs blueshift with the increase of $D$, and the maximum amount of blueshift is 900 nm at $\theta$ is 90. We can find that the maximum contrast ratio is 14 dB when $D$ is 90 nm in Figure 5b. By optimizing the geometric parameters of the structure, we obtain the transmittance of the transmission peak and the contrast ratio of the optical switch under the different conditions, which is more widely used in different applications. It can be seen from the figure that when there is a certain distance between the two structures, the coupling effect between the two structures is better, resulting in higher transmittance. Generally speaking, the closer the distance between the two holes, the stronger the coupling between the excited resonance modes. But in the hoof-shaped structure, when there is a certain distance between the U-shaped structure and the rectangle structure, the metal part in the middle can be regarded as a metal nanoparticle, that is, an electric dipole. When the incident light is irradiated, the interaction between photons and electrons will be greatly enhanced due to the existence of the electric dipole [33]. Therefore, in a certain range, the larger the distance between the U-shaped structure and the rectangle structure, the larger the size of the metal nanoparticles in the middle, the better the resonance coupling effect, that is, the greater the transmission. In addition, according to the relationship between the structure size, transmittance and the contrast ratio, the optimal range of $D$ is 40–80 nm.

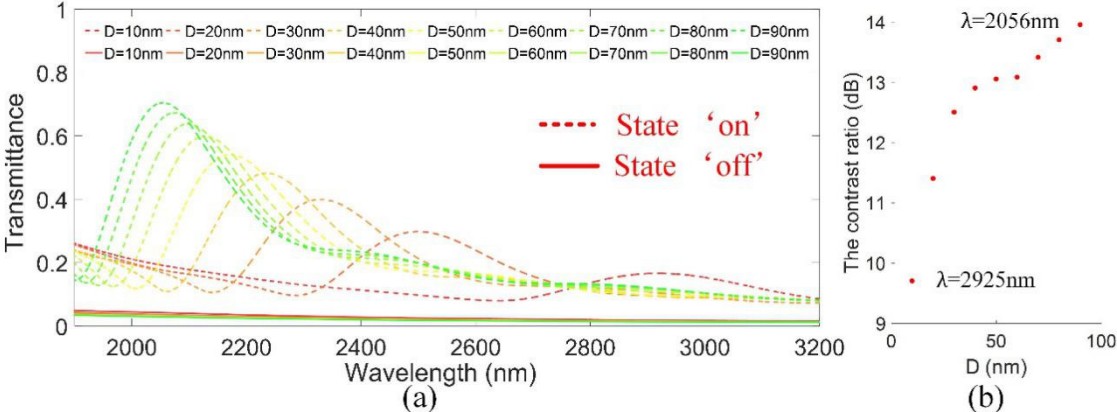

**Figure 5.** (**a**) Transmittance spectrum of changing the distance *D* of hoof-shaped structure; (**b**) The contrast ratio and working wavelength of plasmonic optical switch under different distance *D*.

### 4.2. Design of Plasmonic Optical Splitter

Based on the polarization sensitivity that causes the transmission peak to split in the near-infrared range, we constructed a plasmonic optical splitter in the near-infrared range with parameters shown in Figure 1b. As shown in Figure 2b, we deem that the center wavelengths of two different transmission peaks in the near-infrared range of the plasmonic optical splitter (PO splitter) are the working wavelengths of the corresponding channels of the PO splitter. At the same time, the intensity splitting ratio (ISR) as a performance index to quantitatively describe the spectroscopic effect of the PO splitter can be defined as [34]:

$$\text{ISR} = \max(T_{\max,1}, T_{\max,2})/\min(T_{\max,1}, T_{\max,2}) \tag{2}$$

where the $T_{\max}$ represents the EOT peak at the central wavelength when the PO splitter is in working state and subscripts 1 and 2 are the channels of PO splitter 1 and 2, respectively.

As shown in Figure 6, we change the angle $\theta$ of the incident light. The parameters are as follows: the polarization angle $\theta$ is 0, 10, 20, 30, 40, 50, 60, 70, 80 and 90; other parameters remain unchanged. In this figure, the green dot patterns are the transmittance of channel 1, which decrease with the augment of angle $\theta$; The red dot patterns are the transmittance of channel 2, which increase with the augment of angle $\theta$. Obviously, the transmittance of the two channels is almost equal at $\theta$ is 40. That is to say, the average intensity can be achieved. More important, the POS (splitter) can achieve the controllable function of the device by changing the angle $\theta$ to control the transmittance of two channels.

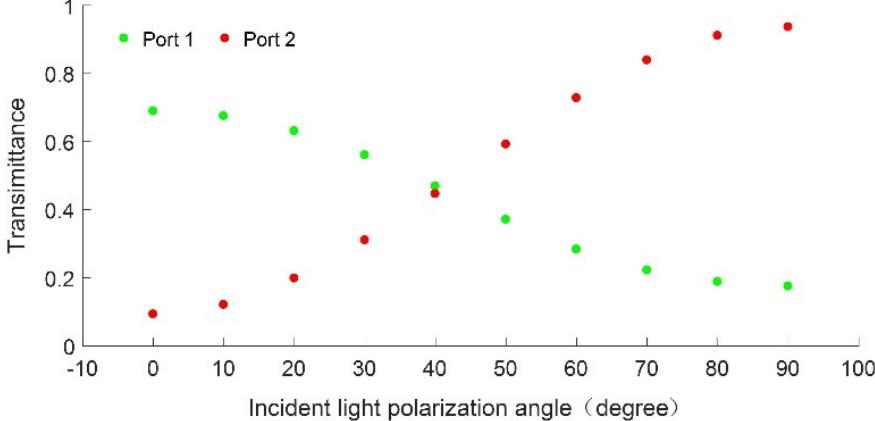

**Figure 6.** The intensities splitting ratio of filter at different $\theta$ with two channels.

## 5. Conclusions

In summary, we investigated the EOT effect via periodic subwavelength hoof-shaped arrays in a metal film. It is found that when the polarization angle of incident light changes, the transmission peak splits in the near-infrared range and a new transmission peak is generated in the mid-infrared range. In addition, wea compare the EOT effects of U-shaped structure, rectangle structure and hoof-shaped structure under different polarization angles. It is found that when the polarization angle changes, the transmission peaks of U-shaped structure and rectangle structure in the near-infrared range interact with each other, and the transmission peak of U-shaped structure in the mid-infrared range occurs redshift.

Under the condition of the realization of EOT effect shown above, we propose a novel plasmonic multifunction device via periodic hoof-shaped structure arrays in difference wavelength ranges. First, by changing the tips length $S_1$ and the separation distance $D$ of the hoof-shaped structure, it is possible to adjust the peaks (amplitude and position) of the PO switch in the mid-infrared range. The performance of the switch can exceed 13.2 dB. Moreover, we can control the transmittance of two channels of the PO splitter by changing the polarization angle. When the angle is 40, the dual channel intensities sharing on average can be realized. In this paper, based on the same structure of different plasmonic optical devices, it can realize the function in different ranges at the same time, and achieve the multiple applications of the device.

**Author Contributions:** Conceptualization, K.W. and J.Y.; data curation, K.W.; methodology, K.W.; software, K.W., Z.Z., X.J. and J.H.; project administration, J.Y.; supervision, J.Y.; writing—Original, K.W.; writing—Review, J.Y.; data analysis, K.W. All authors have read and agreed to the published version of the manuscript.

**Funding:** This work was supported by the National Natural Science Foundation of China (60907003, 61805278), the China Postdoctoral Science Foundation (2018M633704), the Foundation of NUDT (JC13-02-13, ZK17-03-01), the Hunan Provincial Natural Science Foundation of China (13JJ3001) and the Program for New Century Excellent Talents in University (NCET-12-0142).

**Conflicts of Interest:** The authors declare no conflict of interest.

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
