# Peer review of "Plasmonics Induced Multifunction Optical Device via Hoof-Shaped Subwavelength Structure"

_applsci, doi:10.3390/app10082713_

Round 1

Reviewer 1 Report

Wen et al. describe a new plasmonic substrate, termed “hoof-shaped,” which they characterize and evaluate using finite-difference time-domain simulations. While it is unclear what the intended purpose or application of this material will be, the characterizations that were performed are well-thought out. I find this content to be suitable for publication in Applied Sciences, pending clarification of a couple minor concerns, detailed below.

77: The authors should detail how their FDTD method was implemented in this section. Specifically, was it through in-house code, or through a commercial FDTD solver (e.g., COMSOL, Lumerical, EM Explorer, FDTD++, etc.)? Additionally, Yee cell size should be provided for determination of the accuracy of the results.

119: Please fully define “POS” as “plasmonic optical switch” in parentheses. The remaining “(switch)” texts may then be removed throughout the rest of the manuscript. As it currently reads, the text is confusing.

148: Please define the distance or range of distances for which there is more efficient coupling.

159: Why did the authors define the ISR parameter, but never use it? If it is defined, it should be applied to the structures in their simulations?

Reviewer 2 Report

This paper concerns the periodic hoof-shaped subwavelength structure. I believe that the obtained results justify publication of the paper in Applied Sciences. However, the authors need to revise the paper with respect to the comments.

1.
The authors show transmittance of u-shaped, rectangle, and hoof-shaped structure array in Fig. 3. Peak II, III are shifted when the structure changes from u-shaped and rectangle structure to hoof-shaped structure. The authors need to describe what physical phenomena influence these peaks shift.

2.
The authors show transmittance of the hoof-shaped structure when S1 and D are changed in Fig. 4(a) and Fig. 5(a). The readership of this paper would like to know why these change happen. Therefore, the author need to describe why changing S1 and D causes such a change in transmittance.

3.
The authors should add axis labels in Fig. 3.

Round 2

Reviewer 2 Report

1. The authors replied about peak shifts(Peak II, III in Fig. 3). The authors explained that the shifts were caused from the coupling effect of charge. However, the readership of this paper could not figure out the phenomenon by this explanation. The authors should describe which parameters affect the shifts. The authors should use these descriptions to explain the phenomenon.

2. The authors replied about the change of transmittance in Fig. 4(a) and Fig. 5(a). In Fig. 4(a), the authors explained that the change of transmittance is due to that change of resonance mode in the U-shaped structure. How does the change of resonance mode affect the transmittance shift?
In Fig. 5(a), the authors explained the coupling effect affects the change of transmittance. How does the coupling effect affect the transmittance shift?

3. Is the label of S (nm) in Fig. 4 (b) mistook for S1 (nm)?

Round 3

Reviewer 2 Report

The authors explained that the U-shaped structure is coupled with the resonance mode (LSP resonance) in the rectangle, and then the peak III moves. The authors need to explain the reason why the peak III moves. Why does the peak III shift from the wavelength of near 1900 nm to the wavelength of near 2200 nm when the single U-shaped structure changes to the hoof-shaped structure? What is the the difference between the electric field distribution of the single U-shape structure at the wavelength of near 1900 nm and that of the hoof-shaped structure at the wavelength of near 2200 nm?

As with peak III, the authors need to explain the reason why the peak II moves. Why does the peak II shift from the wavelength of near 1600 nm to the wavelength of near 1400 nm when the rectangle structure changes to the hoof-shaped structure? What is the the difference between the electric field distribution of the rectangle structure at the wavelength of near 1600 nm and that of the hoof-shaped structure at the wavelength of near 1400 nm?

Author Response

This manuscript is a resubmission of an earlier submission. The following is a list of the peer review reports and author responses from that submission.

Round 1

Reviewer 1 Report

In the revised version of the manuscript, the authors explain the transmission spectra of the hoof-shaped nanostructure by electromagnetic coupling of the plasmon modes excited on the rectangular holes with the ones excited on the U-shaped holes. As a result of couplings between the peaks I and II, and between the peaks II and III, peak I is blueshifted and peak III is redshifted. This scenario makes sense, however, as I mentioned before, it needs further evidence. CMT calculations are already adopted from the transmission spectra so, it is not further evidence. The authors would have calculated the transmission spectra of the hoof-shaped nanostructure where the individual holes are well away from each other so that they are no longer coupled. If authors’ explanation is right, then, in this case one expects that all three peaks appear without shifting from the spectral locations of their individual spectra. In this form, the explanation is nothing but a speculation. Furthermore, it was very hard for me to catch the coupling scenario from the revised text. I needed to read both the response from the authors and the revised manuscript couple of times to have a clue about what was said. The potential readers of this paper will not have the chance to read the response letter. The ambiguity arose from the wrong emphases and focuses in the text. Unless the sample geometry is symmetric, changing incident polarization changes the transmission spectra of any geometry. Incident light aligns the free charges on the metal along its polarization direction. If the polarization direction is changed from x to y, the plasmon mode arising from the free charge alignment along the x direction is not be observed any longer, but instead, the plasmon mode arising from the free charge alignment along the y direction is excited. Since the geometry is not symmetric, these plasmon modes correspond to different wavelengths. (And this valid for any geometry that is not symmetric.) That is why the 0- and 90-degree polarized light in this work give rise to different peaks (I and III). Besides, it is never possible to shift the plasmon resonance of a passive nanostructure by changing only the polarization unless there is hybridization. Therefore, the statements given in lines 113-116 are misleading in the sense that they overemphasize the obvious as if it is significant. These are the basics of localized plasmon resonances. The authors elaborate a lot on the parts where they explain the effect of changing the incident polarization, instead of providing evidence to the hybridization phenomenon.

The trouble with the given resonance frequency values used in the CMT calculations is still there. The SI unit of angular frequency is rad/s and that of frequency is Hz. Not to mention, Hz and rad/s are not the same units, they are differ by a factor of 2π radians. The resonance values given in lines 141 and 142 are in units of rad/s and correspond to very long wavelengths in the mid-IR region. One can convert the angular frequency to wavelength using ω=2πc/λ. The authors should double check the values.

The authors did not provide any response to my other comments regarding the CMT calculations, nor did they revise the manuscript accordingly.

The optical switching and splitting operations could be done simultaneously with previously examined nanostructures too, as these operations are based on the same phenomena: plasmon hybridization, which is not new and I do not think that this work contributes to the current knowledge.

Although some details about the simulations in Fig. 2 are given in the updated manuscript, the most important detail is still missing. I previously commented that: ‘It is not stated at which wavelengths the field profiles were simulated.’ This means what was the wavelength of the transmitted light that was collected at the detection plane. It is very important because there are multiple resonances and it is not possible to obtain the information of how the field is distributed over the near field of the sample for all resonances at once. The authors state in the response letter that the color bar in Fig. 2 shows the electric field intensity of the detect light divided by the electric field intensity of the incident light but it is still in units of electric field (V/m) in the manuscript. I previously commented that it is not stated which panel is for which angle of incidence in Fig. 2a and it is still not stated.

The authors did not provide any response to my comment: ‘It does not make any sense how changes in the orders of nanometres in the nanostructure geometry could lead to shifts of the resonances in the orders of tens or even hundreds of nanometres.’ Instead they explained why they wanted to observe the spectra while changing these parameters.

Reviewer 2 Report

the description of a  linkage to similar analyses would be beneficial

there is no term plasmon polaron (at the beginning)

instead the term plasmon polariton should be used

(polaron is related with phonons)

pieces in red look strange